# Exploring the Histopathological Features of Thrombus-Associated Localized Amyloid Deposition: Comprehensive Analysis Employing Immunohistochemistry and Proteomics

**DOI:** 10.3390/ijms26104505

**Published:** 2025-05-08

**Authors:** Shojiro Ichimata, Tsuneaki Yoshinaga, Mitsuto Sato, Nagaaki Katoh, Fuyuki Kametani, Masahide Yazaki, Yoshiki Sekijim, Yukiko Hata, Naoki Nishida

**Affiliations:** 1Department of Legal Medicine, Faculty of Medicine, University of Toyama, Toyama 930-0194, Japan; 2Department of Medicine (Neurology and Rheumatology), Shinshu University School of Medicine, Matsumoto 390-8621, Japan; kiccho828@gmail.com (T.Y.);; 3Department of Brain and Neurosciences, Tokyo Metropolitan Institute of Medical Science, Tokyo 156-8506, Japan; kametani-fy@igakuken.or.jp; 4Clinical Laboratory Sciences Division, Shinshu University Graduate School of Medicine, Shinshu University, Matsumoto 390-8621, Japan; mayazaki@shinshu-u.ac.jp; 5Institute for Biomedical Sciences, Shinshu University, Matsumoto 390-8621, Japan

**Keywords:** apolipoprotein A-I, atherosclerosis, lactoferrin, localized amyloidosis, phagocyte, thrombus, proteomics, transthyretin

## Abstract

Amyloid deposition has been reported to localize within thrombi; however, its pathological characteristics, particularly its precursor proteins, remain poorly understood. This study aimed to elucidate the pathological features of thrombus-associated amyloid deposition by immunohistochemistry combined with proteomic analyses using liquid chromatography–tandem mass spectrometry with laser microdissection. Our findings revealed that thrombus-associated amyloid deposits within the thrombus and vessel wall primarily comprised apolipoprotein A-I, with a mixture of amyloid fibrils derived from amyloidogenic proteins, including transthyretin and lactoferrin. Given that these proteins are present in the blood, our results support a previous hypothesis that proteins denatured during thrombus aging are a source of amyloid. Furthermore, phagocytes were infiltrated around the intramural and extravascular deposits rather than around the amyloid deposits within the thrombus. Therefore, amyloid deposits generated within the thrombus may be transported from regions with limited blood flow to the vessel wall and surrounding tissues, where blood flow is present, during thrombus processing. These deposits were primarily removed by phagocytic cells. Our results suggest that a facilitative effect on deposition occurs via a cross-seeding mechanism between amyloid fibrils and that phagocytes can remove amyloid deposits. These findings help elucidate the pathogenesis of localized amyloidosis.

## 1. Introduction

Amyloidosis is characterized by extracellular deposition of misfolded, insoluble amyloid fibrils, which are oriented in a beta-sheet structure [1]. It is classified according to the type of amyloid precursor protein, the pattern of deposition (systemic or localized), and whether it is hereditary or acquired [1]. Localized amyloid deposits occur not only in various organs associated with certain disorders but also in the same tissues as part of normal aging [2]. Furthermore, localized amyloid deposition may function as a waste container (“wasteosome”) [3,4]. Thus, distinct mechanisms may underlie the formation, degradation, and elimination of localized amyloid deposits under different conditions. Notably, Castellani et al. reported an autopsy case of giant cell vasculitis, a localized amyloidosis occurring in the central nervous system, as an adverse reaction to antibody therapy for amyloid-beta [5]. Thus, histiocytes may be involved in the removal of localized amyloid deposits, and elucidating the pathogenesis of localized amyloidosis could contribute to advancements in amyloid removal therapy. However, compared with systemic amyloidosis, localized amyloidosis outside the central nervous system remains insufficiently explored, probably because it rarely causes severe functional impairment in the affected organs.

Localized amyloid deposition occurs in cardiovascular organs. Age-related amyloid deposition, including lactadherin (medin)-derived and apolipoprotein A-I-derived amyloid (AApoAI), takes place in large arteries [6,7,8,9]. Additionally, amyloid deposition has been reportedly associated with atherosclerosis and thrombotic materials in cardiovascular organs [10]. These amyloid deposits are resistant to permanganate treatment, with their deposition pattern suggested to originate from components within the thrombus [10]. However, to the best of our knowledge, the precursor proteins of thrombus-associated amyloid deposition have not yet been analyzed using immunohistochemistry (IHC) based on proteomic analysis using laser microdissection with liquid chromatography–tandem mass spectrometry (LMD with LC-MS/MS).

Hence, this study first examined the frequency and extent of amyloid deposition associated with thrombus formation, followed by conducting proteomic analysis on two patients with high amyloid deposition and sufficient sample volume. Finally, on the basis of these results, IHC analysis was conducted on patients with some degree of amyloid deposition and the findings were evaluated.

## 2. Results

### 2.1. General Appearance of Thrombus-Associated Amyloid Deposition

From the archives of patients with medicolegal autopsy in our department, six aortic aneurysms with thrombi; seven and eleven cases of internal carotid and coronary artery stenosis, respectively, with fibroatheroma showing hemorrhage and thrombus formation; two true brain aneurysms with thrombi; pulmonary and leg venous thrombi from six cases of pulmonary thromboembolism; and four other large artery aneurysm cases with thrombi were selected. Table 1 summarizes the clinical information and the frequency of amyloid deposition in these cases.

As previously reported [10], amyloid deposition occurred in both less organized and hyalinized thrombi, as well as in the area between the sclerotic replacement of the thrombus and the vessel wall. However, it was not observed in the intracranial, pulmonary, and lower extremity vessels, where atherosclerosis was not prominent. Apart from thrombus-associated amyloid deposition, four patients had systemic sporadic transthyretin-derived amyloid (ATTR) deposition, and none had a medical history of familial amyloidosis. Table 2 summarizes the histological findings at each deposition site, whereas Figure 1 and Figure 2 show the representative micrographs of thrombus-associated amyloid deposition in the aorta and other large arteries, respectively.

Nineteen specimens from 18 cases had amyloid deposition, which was located solely in the intrathrombus region in 4 cases, in the border to the intravascular wall region in 3 cases, and in both regions in 12 cases. Generally, amyloid deposition was more severe in the border area to the vessel wall than within the thrombus, but the difference was not significant (*p* = 0.39). Furthermore, six cases demonstrated phagocytosis of the deposits by histiocytes and multinucleated giant cells.

### 2.2. Relationship Between Thrombus-Associated Amyloid Deposition and Phagocytes

In cases of amyloid deposit phagocytosis, the relationship between amyloid deposition and phagocytes was further examined. Figure 3 presents representative photographs showing the association between thrombus-associated amyloid deposits and phagocytes.

Interestingly, little or no phagocyte infiltration was observed around the amyloid deposits in the thrombus (Figure 3a–c), but it was evident around the amyloid deposits within the vessel wall (Figure 3d). Furthermore, granulomas formed in the perivascular area, and macrophages and multinucleated giant cells phagocytosed fragmented amyloid deposits in the area (Figure 3e,f). Figure 4 shows a case with findings showing an interesting association between amyloid deposits and phagocytes.

In this case, amyloids are deposited in the vessel wall, both with and without surrounding histiocyte or multinucleated giant cell infiltration (Figure 4a). Interestingly, under polarized light, the intensity of apple-green birefringence was lower in the deposits associated with phagocytic cells than in those without, suggesting the inflammation-induced degradation of amyloid fibrils (Figure 4b).

### 2.3. Proteomic and Immunohistochemical Features of Thrombus-Associated Amyloid Deposition

First, in a representative case (Case 18), IHC using antibodies employed in the Japanese amyloidosis consultation system was attempted [11], but no clear immunoreactivity was observed. Therefore, proteomic analysis using LMD with LC-MS/MS, as previously described [12,13], was conducted. Figure 5 summarizes the analyses for Cases 18 and 19.

In addition to several amyloid-associated proteins, four proteins with potential amyloidogenic properties were detected (Figure 5a,b). Both cases showed no lactadherin (medin). The presence of these deposits in pathological specimens was confirmed by IHC performed on cases with some degree of amyloid deposition (Grade ≥ 2+) using commercially available monoclonal antibodies previously reported to show positivity in amyloid deposits [4,14,15,16,17]. Figure 6 shows the representative IHC findings, while Table 3 summarizes the analysis results.

All cases were mainly positive of ApoAI, followed by lactoferrin (75%/88%) and transthyretin (50%/50%), in both intrathrombus and border-to-intramural amyloids. No case showed immunoreactivity for Igκ. However, in all cases, the immunoreactivity in the thrombus-associated amyloid deposits was weak to moderate and lacked uniformity (Figure 6a–f), with no areas exhibiting strong immunoreactivity throughout the deposits, as observed in the systemic ATTR deposition foci (Figure 6e, Case 1). Furthermore, all examined organs, including the kidneys, of all cases showed no finding suggestive of hereditary AApoAI amyloidosis complications [18]. These results suggest that these localized amyloid deposits are primarily composed of AApoAI, with a mixture of amyloid fibrils derived from amyloidogenic proteins circulating in the blood.

## 3. Discussion

This study demonstrated that thrombus-associated amyloid deposits are primarily composed of AApoAI, with a mixture of amyloid fibrils derived from circulating amyloidogenic proteins. These findings align with previous reports of AApoAI deposition in atherosclerotic foci of the aorta and internal carotid artery, suggesting that these lesions also originate from ApoAI in the blood [6,7,19]. The dissociation of native TTR tetramers into monomers leads to amyloid fibril formation [20]. Thus, during thrombus formation and its subsequent degeneration, structural changes in amyloidogenic proteins in the blood may have contributed to amyloid formation. This supports the hypothesis proposed by Goffin et al., i.e., certain thrombus components undergo transformation into amyloid fibrils during the aging process of the thrombi [10]. Notably, in the present study, amyloid formation within the thrombus was observed only in vessels with atherosclerosis, strongly suggesting that atherosclerosis plays a role in its development. Considering this pathology, amyloid deposition might be intensified by recurrent thrombus formation caused by atherosclerosis [21], making it detectable histopathologically. The impact of amyloid formation within the thrombus remains unclear, but it may contribute to thrombus instability. Hence, further investigation is needed. Of note, although Goffin et al. provided a detailed histopathological distribution of amyloids where the precursor proteins were not identified, and to the best of our knowledge, this amyloid deposition has not yet been analyzed in detail in nearly 40 years [10]. As previously reported, amyloid precursor proteins that are difficult to determine using histopathological methods alone can be identified by combining IHC and proteomics using LMD with LC-MS/MS [22,23].

Interestingly, ATTR and AApoAI have been reported to be codeposited in musculoskeletal diseases such as spinal canal stenosis and knee osteoarthritis [9,14,24]. Our findings suggest that amyloid deposition at these sites may result from precursor proteins leaking from blood vessels and undergoing structural changes resulting from mechanical stress and inflammation. Notably, phagocytic infiltration around the amyloid deposits is not evident in the micrographs in these previous reports, which is consistent with our observations. This finding is particularly interesting because it differs from localized immunoglobulin light chain-derived amyloidosis, in which multinucleated giant cells may play a crucial role [25]. These findings support previous reports suggesting a cross-seeding mechanism between AApoAI and ATTR, where each promotes the deposition of the other [14]. The effects of these amyloid deposits on musculoskeletal tissues and thrombi, particularly regarding tissue damage and thrombus resolution, remain unclear, thereby warranting further investigation.

Unlike intrathrombus amyloid deposition foci, phagocytes were observed surrounding the amyloid in the vessel wall. Additionally, granulomas were noted forming in the perivascular area of a very old thrombus-occluded aneurysm, as well as amyloid fragments phagocytosing in the foci. In recent years, several reports have implicated inflammatory cell infiltration with multinucleated giant cells in removing amyloid deposits [26,27,28]. On the basis of our findings and the results of previous reports [10], amyloids generated within the thrombus might undergo the following processes: First, amyloids generated within the thrombus resist digestive enzymes during thrombus organization and become trapped in dense sclerotic appositional tissue. At this stage, the blood flow to the amyloid deposits is nearly blocked, preventing them from accessing the deposits; hence, slight inflammation or phagocytosis occurs. The amyloids are then gradually degraded and absorbed by phagocytes from the vasa vasorum and/or recanalized vessels resulting from the organization of the surrounding thrombotic tissue. The unabsorbed portion is transported to the perivascular area, where blood flow is more abundant, and is processed by more phagocytes. Figure 7 illustrates this hypothesis.

Of note, progressive degradation and resorption associated with inflammation may obscure amyloid deposition in pathological specimens [28]. Under polarized light, reduced birefringence in amyloid deposits with surrounding phagocytic infiltration was noticed, and some lesions could not be histologically confirmed as amyloid. In thrombus-associated amyloid deposition, the immunoreactivity was not uniform and varied between the deposition areas. Similarly, previous histopathological studies of spinal canal stenosis and knee osteoarthritis have reported cases wherein amyloid could not be typed [14], possibly because of degeneration. Future research is needed to enhance the sensitivity of histopathological detection in such cases.

This study has some limitations. In addition to potential biases in our study population, it has a small number of cases with moderate or greater amyloid deposition. Furthermore, our laboratory lacked a reliable antibody against the variable region of Igκ, which likely prevented amyloid detection by IHC. Additionally, only a few cases exhibited moderate or greater deposition with sufficient sample quantity, allowing proteomic analysis to be performed in only two cases. The *ApoAI* and *TTR* genes were also not analyzed. Therefore, the possibility of hereditary amyloid deposition associated with these mutations could not be ruled out.

## 4. Materials and Methods

### 4.1. Subjects

The autopsy records in our laboratory documented between 2009 and 2023 were retrospectively examined. Data on patients’ demographic and clinical characteristics (including the cause of death) were retrieved from the medical records of police examinations and contributions from family members or from the primary physician if a record indicated clinic visits.

### 4.2. Tissue Samples

Specimens were fixed in 20% buffered formalin and routinely embedded in paraffin. Then, they were sectioned at 4 μm thick, followed by hematoxylin–eosin staining, elastica–Masson staining, or IHC. Furthermore, 6 μm thick sections were prepared for phenol Congo red (pCR) staining [29].

### 4.3. Semiquantitative Grading System for Thrombus- and Atheromatous Plaque-Associated Amyloid Deposition

Following amyloid deposition confirmation by pCR staining, the severity of the amyloid deposition in the thrombus or fibroatheroma with hemorrhage and the area between the sclerotic replacement of the thrombus to the adventitia of the vessel were assessed semiquantitatively. For this evaluation, given the absence of previous reports assessing the severity of thrombus-related amyloid deposition, a new 4-point scoring system was developed based on our previous study [30] and applied as follows:Grade 0, none;Grade 1+, focal and tiny amyloid deposition;Grade 2+, multifocal deposition with nodular deposits; andGrade 3+, multifocal deposition with some massive deposits.

Figure 8 shows representative microphotographs illustrating the deposition patterns of this grading system.

Furthermore, the intensity of immunoreactivity in the amyloid deposits was assessed using a 4-point scoring system as follows: Grade 0, negative; Grade 1+, weak; Grade 2+, moderate; and Grade 3+, strong. In this evaluation system, the intensity of immunoreactivity in the small vessels of patients with systemic ATTR was defined as Grade 3+ and compared with that shown in Figure 6e’s inset.

### 4.4. IHC

Deparaffinized sections were treated with 3% H_2_O_2_ for 15 min to inactivate endogenous peroxidase, immersed in 98% formic acid for 1 min for antigen retrieval, and subjected to immunostaining. For IHC, primary antibodies against ApoAI (rabbit, clone EP1368Y, 1:4000; Abcam, Cambridge, UK), Igκ (rabbit, clone H16-E, 1:500; DB Biotech, Kosice, Slovakia), lactoferrin (mouse, clone B97, 1:150; Santa Cruz, TX, USA), and prealbumin (TTR) (rabbit, clone EPR3219, 1:2000; Abcam) were used. Table 4 summarizes the details of the IHC methods.

### 4.5. Proteomics Analysis Using Mass Spectrometry

Amyloid deposition in the thrombus and in the border area to the vessel wall was analyzed using LMD followed by LC-MS/MS. Sections of paraffin-embedded samples, 6 μm thick, were placed on slide glasses with a special foil for LMD (Leica Microsystems Inc., Tokyo, Japan) and subsequently stained with Congo red. Next, the microdissected tissues were solubilized in 50 mL of 10 mM Tris/1 mM EDTA/0.002% Zwittergent 3–16 (Calbiochem, San Diego, CA, USA) buffer with heating at 98 °C for 90 min and sonication for 60 min, followed by digestion with trypsin overnight at 37 °C. After adding 2 μL of 100 mM DTT, the mixture was incubated at 100 °C for 5 min, followed by drying and storage at −80 °C until assay. Thereafter, the digests were resuspended in 0.1% formic acid and introduced into a nanoflow HPLC system (EASY-nLC 1200, Thermo Fisher Scientific Inc., Waltham, MA, USA). A packed nanocapillary column NTCC-360/75-3-123 (0.075 mm I.D. × 125 mm L, particle diameter 3 μm, Nikkyo Technos Co., Ltd., Tokyo, Japan) was used at a flow rate of 300 nL/min with a 2–80% linear gradient of acetonitrile for 80 min. Eluted peptides were directly detected using an ion trap mass spectrometer (QExactive HF; Thermo Fisher Scientific Inc., Waltham, MA, USA). For ionization, a spray voltage of 2.0 kV and capillary temperature of 250 °C were applied. The mass acquisition method included one full MS survey scan (Orbitrap resolution, 60,000) and an MS/MS scan of the most abundant precursor ions from the survey scan (Orbitrap resolution, 15,000). Dynamic exclusion for the MS/MS was set to 30 s. An MS scan range of 350–1800 m/z was employed in the positive ion mode, followed by data-dependent MS/MS using the HCD operating mode on the top 15 ions in order of abundance. The data were analyzed using Proteome Discoverer (Thermo Fisher Scientific Inc., Waltham, MA, USA), Mascot software version 3.1 (Matrix Science Inc., Boston, MA, USA), and Scaffold DDA software version 6.5.0 (Proteome Software, Inc., Portland, OR, USA). Swissprot and GenBank databases were used [12,13].

### 4.6. Statistical Analysis

Data were analyzed using IBM SPSS statistics version 29 (SPSS Inc., Chicago, IL, USA). A two-tailed *p*-value below 0.05 was considered significant. Ordinal variables (pathological scores) were compared using the Mann–Whitney *U* test.

## 5. Conclusions

This study demonstrated the immunohistochemical characteristics of thrombus-associated localized amyloid deposition according to proteomic analysis using LMD with LC-MS/MS. To the best of our knowledge, this study is the first to reveal that thrombus-associated amyloids are composed of multiple amyloidogenic proteins. This combined approach is highly effective for identifying precursor proteins in previously uncharacterized amyloid deposits in histopathological specimens. Further research on localized amyloidosis is warranted, considering that precursor protein typing may enhance our understanding of amyloidogenesis. Additionally, phagocytic cells may play a key role in removing localized amyloid deposits. Clarifying their function is necessary to improve the efficacy of amyloid removal therapies, particularly for localized amyloids such as amyloid-β in which phagocytes are associated with amyloid removal [5].

## Figures and Tables

**Figure 1 ijms-26-04505-f001:**
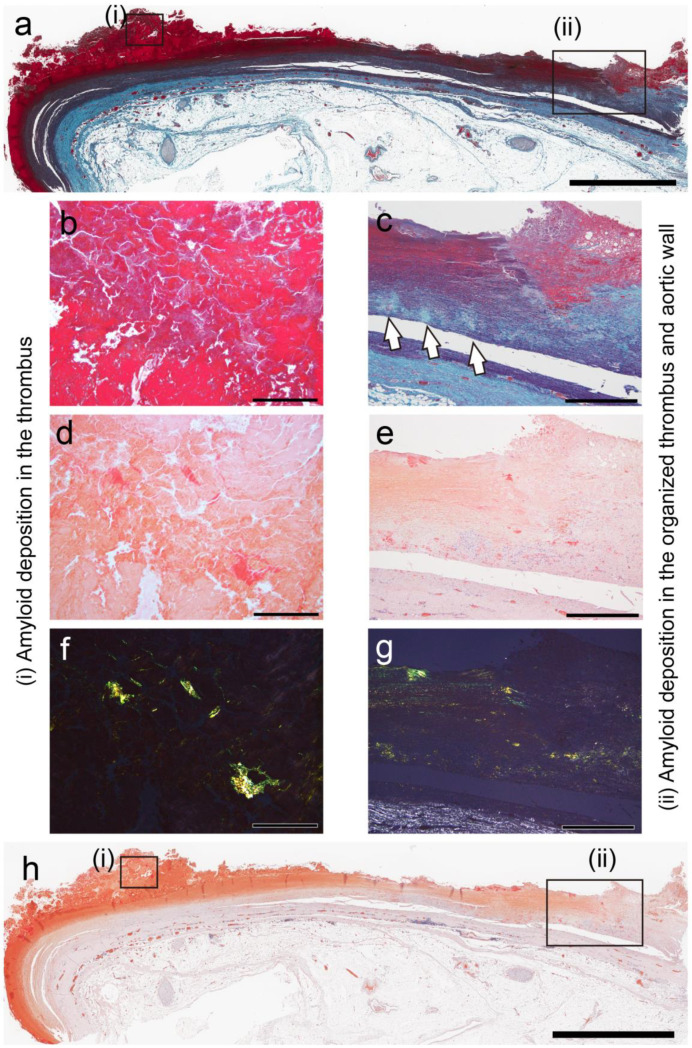
Representative pathological findings in aortic aneurysm (Case 2 in Table 2). (**a**–**c**) Elastica–Masson staining; (**d**–**h**) phenol Congo red (pCR) staining under bright-field (**d**,**e**,**h**) and polarized light (**f**,**g**) observation. (i) Panel a indicates intrathrombus amyloid deposition, whereas (ii) indicates intramural amyloid deposition. Medial elastic fibers are disrupted at amyloid deposition site within wall ((**c**), arrow). Scale bar = 5 mm (**a**,**h**); 2 mm (**c**,**e**,**g**); 200 μm (**b**,**d**,**f**).

**Figure 2 ijms-26-04505-f002:**
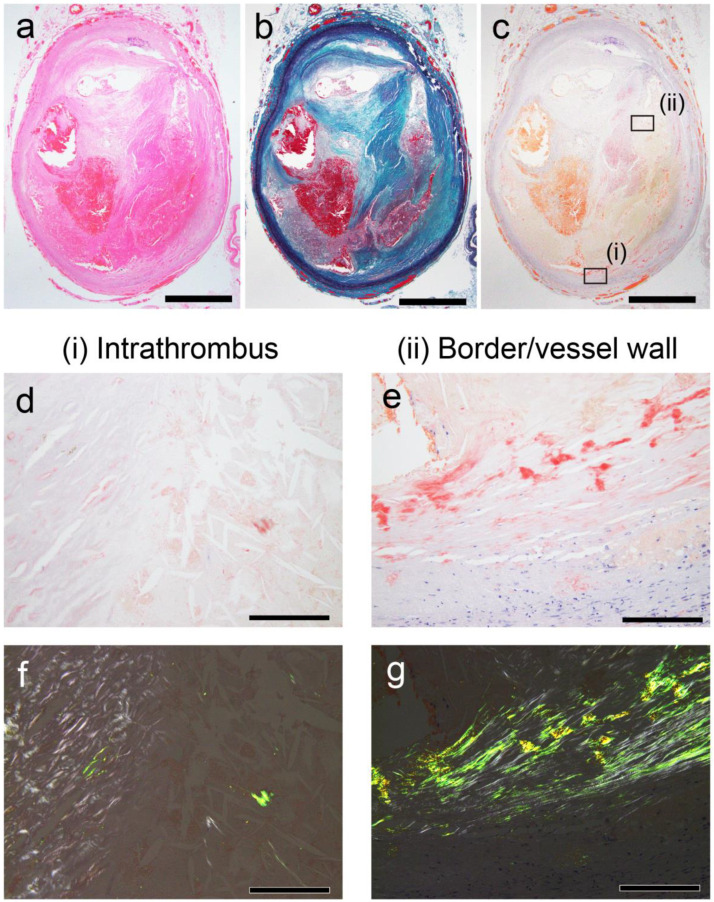
Representative pathological findings in internal carotid artery (Case 8 in Table 2). (**a**) Hematoxylin–eosin staining; (**b**) Elastica–Masson staining; (**c**–**g**) pCR staining under bright-field (**c**–**e**) and polarized light (**f**,**g**) observation. (**a**–**c**) FHT formation is observed. (**d**–**g**) Intrathrombus area and border/vessel wall area contain amyloid deposition. Scale bar = 2 mm (**a**–**c**); 200 μm (**d**–**g**).

**Figure 3 ijms-26-04505-f003:**
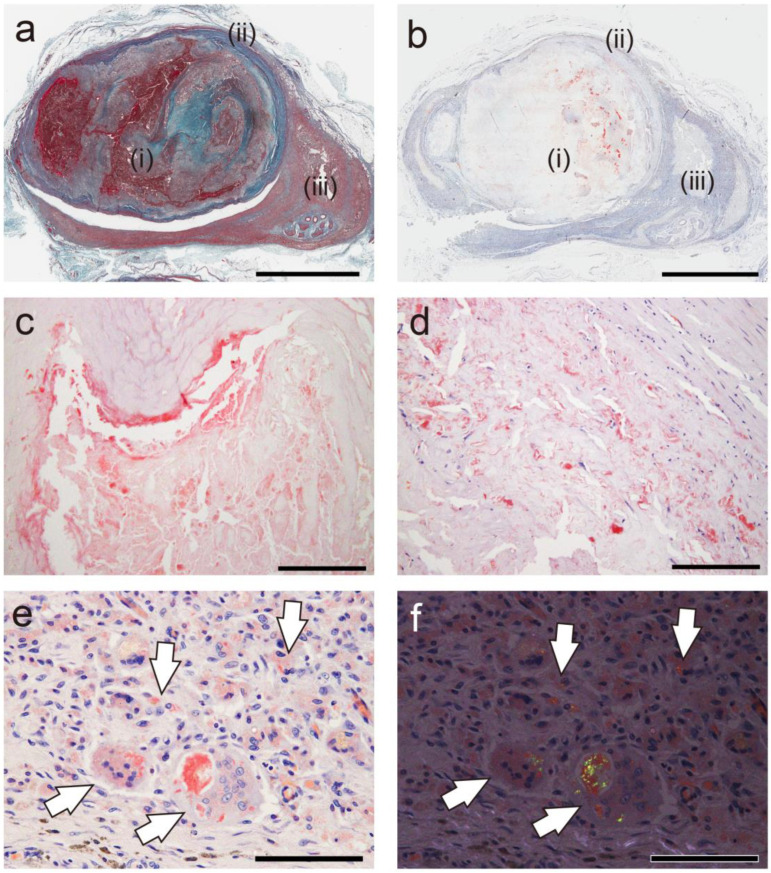
Representative pathological findings of arteriovenous fistula-related aneurysm in forearm approximately 20 years after thrombus occlusion (Case 18). (**a**) Elastica–Masson staining; (**b**–**e**) pCR staining under bright-field (**b**–**e**) and polarized light (**f**) observation. (**a**,**b**) Amyloid deposition is observed in thrombus (i), border area to vessel wall (ii), and granuloma area (iii). Panels (**c**–**e**) are higher-magnification views of amyloid deposition foci in (i), (ii), and (iii), respectively. Notably, phagocyte infiltration is inevident around amyloid deposits in thrombus (**c**), but it is observed within vessel wall (**d**). (**e**,**f**) Amyloid deposition is identified within cytoplasm of macrophages and multinucleated giant cells in granuloma area (arrows). Scale bar = 5 mm (**a**,**b**); 200 μm (**c**,**d**); 100 μm (**e**,**f**).

**Figure 4 ijms-26-04505-f004:**
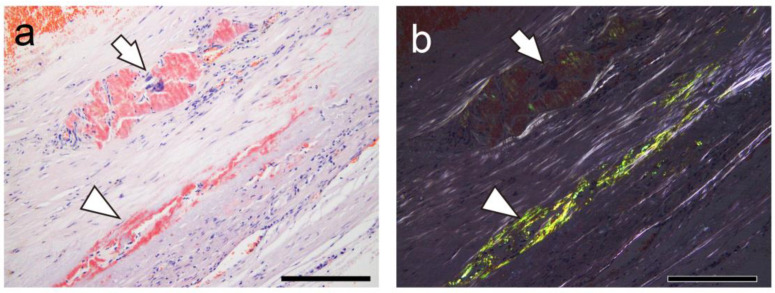
Representative pathological findings showing congophilic state at varying degrees among amyloid deposits (Case 4). (**a**,**b**) pCR staining under bright-field (**a**) and polarized light (**b**) observation. (**a**) Within arterial wall, amyloid deposits are present, one area showing prominent inflammatory cell infiltrate, including multinucleated giant cells (arrow), and another area exhibiting less phagocyte infiltration (arrowhead). Under bright-field observation, congophilic state did not clearly differ between two lesions. (**b**) Under polarized light, apple-green birefringence was weak in former deposition area but was clear in latter. Scale bar = 200 μm (**a**,**b**).

**Figure 5 ijms-26-04505-f005:**
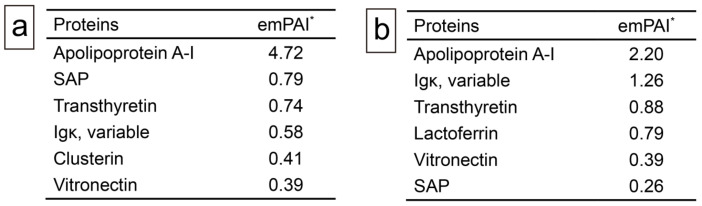
Representative proteomics results. (**a**) Proteins identified in intrathrombus amyloid deposition in Case 19; (**b**) proteins identified in amyloid deposition found in border to vessel wall in Case 18. * emPAI stands for exponentially modified protein abundance index used to estimate relative protein quantification in mass spectrometry-based proteomics analysis. Igκ, immunoglobulin kappa light chain; SAP, serum amyloid P-component.

**Figure 6 ijms-26-04505-f006:**
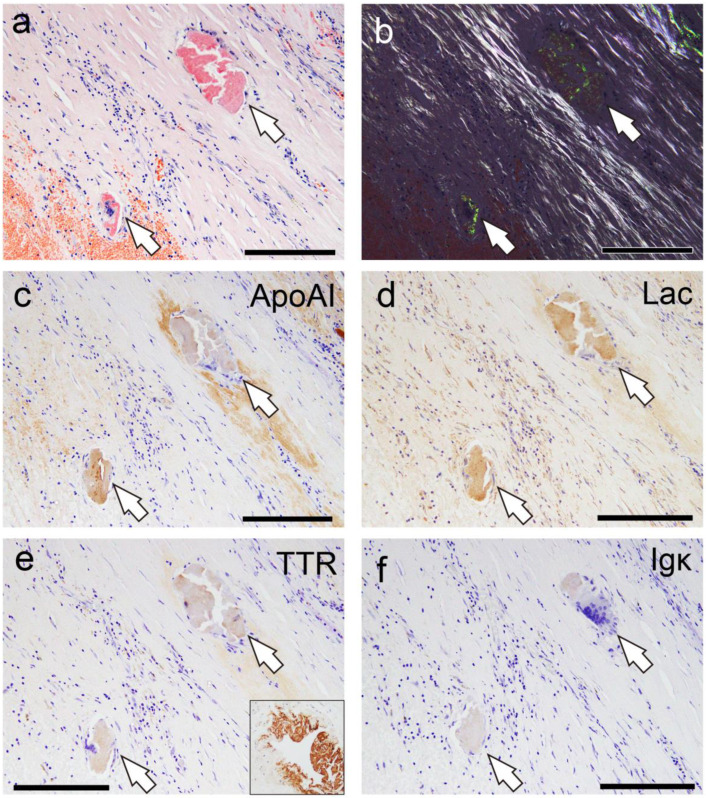
Representative IHC findings in amyloid deposits (Case 4). (**a**,**b**) pCR staining under bright-field (**a**) and polarized light (**b**) observation. IHC for apolipoprotein A-I (ApoAI, (**c**)); lactoferrin (Lac, (**d**)); transthyretin (TTR, (**e**)); and immunoglobulin κ light chain (Igκ, (**f**)). Arrows indicate the amyloid deposits. Amyloid deposits in this case (**a**,**b**) are moderately positive for ApoAI (**c**) and Lac (**d**), weakly positive for TTR (**e**), and negative for Igκ (**f**). (**e**) TTR immunoreactivity in thrombus-associated amyloid deposits is significantly weaker than that observed in patients with systemic ATTR amyloidosis (Case 1) that underwent IHC at the same time as the control with IHC-grade 3+ (inset). Scale bar = 200 μm (**a**–**f**).

**Figure 7 ijms-26-04505-f007:**
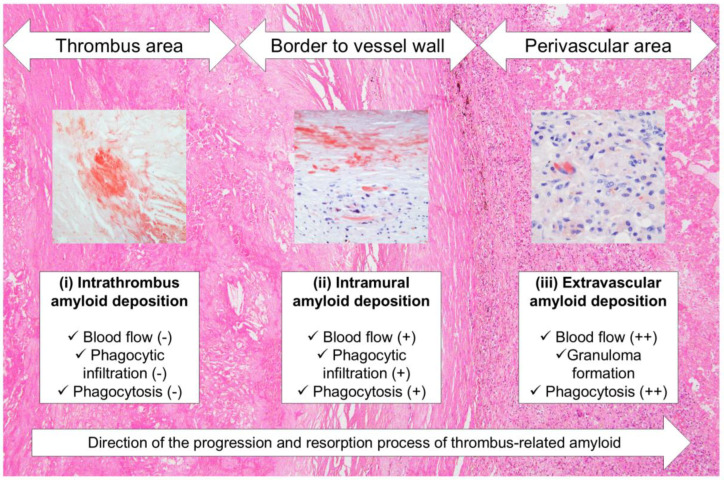
Schematic representation of hypothesized progression and resorption process of thrombus-associated amyloids. (−) Absent; (+) Present to a mild degree; (++) Abundantly present.

**Figure 8 ijms-26-04505-f008:**
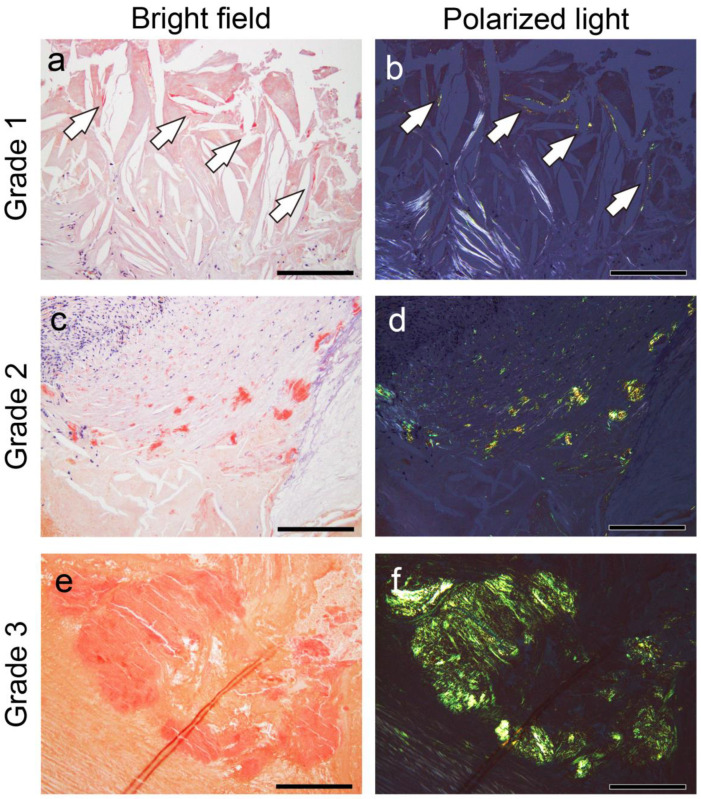
Representative microphotographs of thrombus-associated amyloid deposition evaluated using grading system employed in this study. (**a**–**f**) pCR staining under bright-field (**a**,**c**,**e**) and polarized light (**b**,**d**,**f**) observation. (**a**,**b**) Grade 1+, focal and tiny amyloid deposition (arrow); (**c**,**d**) Grade 2+, multifocal deposition with nodular deposits; (**e**,**f**) Grade 3+, multifocal deposition with some massive deposits. Scale bar = 200 μm (**a**–**f**).

**Table 1 ijms-26-04505-t001:** Summary of evaluated cases.

	Aorta *	CaA	CoA **	CrA **	PA/VLE	Other ***
Age (range)	80.2 ± 8.8 (71–97)	79.0 ± 4.9 (72–88)	72.2 ± 12.8 (47–93)	39.0 ± 8.0 (31–47)	49.3 ± 20.4 (22–81)	76.5 ± 17.8 (47–92)
Sex (F/M)	3/3	1/6	1/10	0/2	4/2	0/4
ATTR-CA-positive (%)	2 (33)	1 (14)	1 (9)	0	0	0
Number of positive cases for amyloid deposition (%)						
Thrombus/FHT	3 (50)	3 (43)	6 (55)	0	0	3 (75)
Border/vessel wall ****	3 (50)	4 (57)	5 (45)	0	0	3 (75)

**Abbreviations:** ATTR-CA, transthyretin cardiac amyloidosis; CaA, carotid artery; CoA, coronary artery; CrA, cranial artery; F, female; FHT, fibroatheroma with hemorrhage and thrombus; M, male; PA, pulmonary artery; VLE, veins of the lower extremities. * Includes aortic aneurysm cases associated with chronic aortic dissection (1 case) and arteriosclerosis (5 cases). ** One case involving three aneurysms: right coronary artery aneurysm, basilar artery aneurysm, and arteriovenous fistula-related aneurysm in forearm. *** Includes aneurysm cases affecting lower mesenteric artery, internal iliac artery, and left common iliac artery and arteriovenous fistula-related aneurysm in forearm. **** Includes area between sclerotic replacement of thrombus and adventitia of vessel.

**Table 2 ijms-26-04505-t002:** Summary of pathological severity in amyloid deposition-positive cases.

Case #	Age	Sex	Anatomical Site	Location and Severity of Amyloid Deposition *		
				Thrombus/FHT	Border/Vessel Wall	Phagocytosis
1	97	F	Abdominal aorta	1+	1+	Positive
2	83	M	Thoracic aorta	3+	3+	Positive
3	81	M	Abdominal aorta	1+	1+	Negative
4	78	M	Abdominal aorta	1+	3+	Positive
5	78	F	Internal carotid artery	0	1+	Negative
6	75	M	Internal carotid artery	1+	1+	Negative
7	77	M	Internal carotid artery	1+	1+	Negative
8	83	M	Internal carotid artery	2+	3+	Positive
9	80	M	Right coronary artery	1+	2+	Negative
10	79	F	Left coronary artery	1+	0	Negative
11	66	M	Left coronary artery	1+	0	Negative
12	73	M	Left coronary artery	1+	2+	Negative
13	72	M	Right coronary artery	1+	0	Negative
14	93	M	Left coronary artery	1+	0	Negative
15 **	47	M	Right coronary artery	0	1+	Negative
16	65	M	Right coronary artery	0	1+	Negative
17	89	M	Internal iliac artery	1+	2+	Negative
18 **	47	M	Arteriovenous fistula	2+	3+	Positive
19	92	M	Common iliac artery	2+	3+	Positive

* Amyloid deposition grading: 0, none; 1+, focal and tiny amyloid deposition; 2+, multifocal deposition with nodular deposits; 3+, multifocal deposition with some large nodular deposits. ** Same patient.

**Table 3 ijms-26-04505-t003:** Summary of immunohistochemical analysis results.

Case #	Severity *	Intrathrombus/FHT Amyloid **				Boder/Vessel Wall Amyloid **			
		ApoAI	Lac	TTR ***	Igκ	ApoA1	Lac	TTR ***	Igκ
2	3+/3+	2+	1+	1+	0	2+	1+	1+	0
4	1+/3+	NE	NE	NE	NE	2+	2+	1+	0
8	2+/3+	2+	1+	0	0	2+	1+	0	0
9	1+/2+	NE	NE	NE	NE	2+	0	0	0
12	1+/2+	NE	NE	NE	NE	1+	1+	0	0
17	1+/2+	NE	NE	NE	NE	1+	1+	0	0
18	2+/3+	2+	1+	0	0	2+	1+	1+	0
19	2+/3+	1+	0	1+	0	2+	1+	1+	0

**Abbreviations:** Lac, lactoferrin; NE, not evaluated; TTR, transthyretin. * Amyloid deposition severity in thrombus/border to vessel wall. ** Immunoreactivity grading: 0, negative; 1+, weak; 2+, moderate; 3+, strong. *** No cases had systemic ATTR amyloidosis.

**Table 4 ijms-26-04505-t004:** Summary of antibodies and IHC methods used in this study.

Antibody	Source	Clone	Dilution	Antigen Retrieval	References
Apolipoprotein A-I	Abcam	EP1368Y	1:4000	98% FA (1 min)	[12]
Immunoglobulin κ light chain	DB Biotech	H16-E	1:500	98% FA (1 min)	[13]
Lactoferrin	Santa Cruz	B97	1:150	98% FA (1 min)	[4,14]
Prealbumin	Abcam	EPR3219	1:2000	98% FA (1 min)	[15]

For immunostaining, the Leica Bond-IV automation and Leica Refine detection kits (Leica Biosystems, Bannockburn, IL, USA) were employed according to the manufacturer’s instructions. Subsequently, all sections were counterstained with hematoxylin.

## Data Availability

The datasets used and analyzed in the current study are available from the corresponding authors upon request.

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
