# Peer review of "Exploring the Histopathological Features of Thrombus-Associated Localized Amyloid Deposition: Comprehensive Analysis Employing Immunohistochemistry and Proteomics"

_ijms, 2025, doi:10.3390/ijms26104505_

Round 1
Reviewer 1 Report
Comments and Suggestions for Authors
In this work, the authors explored the histopathological features of thrombus-associ-ated localized amyloid deposition in the means of immunohistochemistry and proteomics. The topic of this work is interesting and the methods used for the investigation are reasonable. The results of this work are useful to understand the pathogenesis of localized amyloidosis.
Here are some suggestions to improve the manuscript:
Abstract: At the beginning, the background of thrombus-associated amyloid deposition should be briefly introduced.
Introduction: This content of this section should be extended because the background of this work is not thoroughly introduced. For instance, the definition and hazard of amyloidosis, systemic amyloidosis and localized amyloidosis, existing studies that have illustrated the formation, degradation, and elimination of localized amyloid deposits.
Avoid the expression of "we..." throughout the manuscript.
Figure 1: Explain what's the mean of (i) and (ii).
Adjust the position of Table 2 because Figures 1 and 2 are based on the information of Table 2.
The authors should carefully check the sequence of results; this section is not well organized and somewhat confusing.
Discussion can be combined into three sections, focusing on the results of immunohistochemistry, the results of proteomics, the potential mechanisms.
The conclusions of this work can be stood alone.
4.3 Semiquantitative Grading System for Thrombus- and Atheromatous-Plaque–Associated Amyloid Deposition: Add a reference for the grading system.
Figure 8: Why the length of scale bar in different figures is inconsistent? Explain what's the mean of the arrows?
For the rest methods, the detailed procedures should be introduced, especially "4.4 IHC" and " 4.5 Proteomics Analysis Using Mass Spectrometry".
Comments on the Quality of English LanguageThe English could be improved to more clearly express the research.
Author Response
Response to Reviewer #1
We wish to thank you for your kind review and comments that have helped us improve our manuscript substantially. In the revised manuscript, the newly added sentences are written in red.
Response:
Thank you for your comments. As per your comment, we have made the following revisions to the manuscript:
- We have briefly introduced the thrombus-associated amyloid deposition in the abstract (Lines 17–18).
- Information related to the study background has been added to the introduction of the main text (Lines 39–40; 46–50; 52–54). The order of the references has also been changed.
- We have deleted “we...” from the manuscript.
- We have explained (i) and (ii) in Figure 1 (Lines 106–107).
- We have adjusted the position of Table 2.
- The sequence of the results has been adjusted. Furthermore, we have divided the results examining the association between amyloid and phagocytes into a new subsection (subsection2).
- We have combined the first and following paragraphs of the discussion, reducing the number of paragraphs by one.
- We have added a new section for the conclusions.
- Given that the assessment of the severity of thrombus-related amyloid deposition has not yet been reported, we made a new 4-point scoring system for this analysis according to our previous study. We have added this information in Lines 289–291.
- We have adjusted the scale bar length and explained arrows in Panels a and b (Line 300).
- We have added information on IHC and proteomics analysis (Lines 309–311; 322–344).
Reviewer 2 Report
Comments and Suggestions for Authors
The authors characterize thrombus-associated amyloid deposits from human patients. Notably they use LCMSMS to identify proteins present in laser microdissected samples.
I found this manuscript interesting and containing new information. My only criticism lies with Figure 5. The proteomics results are shown for just two Cases. Since the application of proteomics is an important step forward I can’t see why the results for just 2 cases are shown. If the authors have the data for all cases I think it should be shown in its entirety. This would make the paper much stronger.
Author Response
Response to Reviewer #2
We wish to thank you for your kind review and comments that have helped us improve our manuscript substantially. In the revised manuscript, the newly added sentences are written in red.
Response:
Thank you for your comments. We could evaluate only two cases because only few exhibited moderate or greater deposition with sufficient sample quantity. This point has been added as one of the study limitations (Lines 255–257).
Round 2
Reviewer 1 Report
Comments and Suggestions for Authors
The revised manuscript is fine.